# Ageing Curtails the Diversity and Functionality of Nascent CD8^+^ T Cell Responses against SARS-CoV-2

**DOI:** 10.3390/vaccines11010154

**Published:** 2023-01-11

**Authors:** Davide Proietto, Beatrice Dallan, Eleonora Gallerani, Valentina Albanese, Sian Llewellyn-Lacey, David A. Price, Victor Appay, Salvatore Pacifico, Antonella Caputo, Francesco Nicoli, Riccardo Gavioli

**Affiliations:** 1Department of Chemical, Pharmaceutical and Agricultural Sciences, University of Ferrara, 44123 Ferrara, Italy; 2Department of Environment and Prevention Sciences, University of Ferrara, 44123 Ferrara, Italy; 3Division of Infection and Immunity, Cardiff University School of Medicine, Cardiff CF14 4XN, UK; 4Systems Immunity Research Institute, Cardiff University School of Medicine, Cardiff CF14 4XN, UK; 5Université de Bordeaux, CNRS UMR 5164, INSERM ERL 1303, ImmunoConcEpT, 33000 Bordeaux, France

**Keywords:** CD8^+^ T cells, immunosenescence, SARS-CoV-2

## Abstract

Age-related changes in the immune system are thought to underlie the vulnerability of elderly individuals to emerging viral diseases, such as coronavirus disease 2019 (COVID-19). In this study, we used a fully validated *in vitro* approach to determine how age impacts the generation of *de novo* CD8^+^ T cell responses against severe acute respiratory syndrome coronavirus-2 (SARS-CoV-2), the causative agent of COVID-19. Our data revealed a generalized deficit in the ability of elderly individuals to prime the differentiation of naïve precursors into effector CD8^+^ T cells defined by the expression of interferon (IFN)-γ and the transcription factor T-bet. As a consequence, there was an age-related decline in the diversity of newly generated CD8^+^ T cell responses targeting a range of typically immunodominant epitopes derived from SARS-CoV-2, accompanied by an overall reduction in the expression frequency of IFN-γ. These findings have potential implications for the development of new strategies to protect the elderly against COVID-19.

## 1. Introduction

In December 2019, a novel coronavirus, severe acute respiratory syndrome coronavirus-2 (SARS-CoV-2), was discovered as the causative agent of an outbreak of lower respiratory tract infections in Wuhan, China. As the virus spread globally, it became clear that elderly individuals responded suboptimally to vaccination and were particularly susceptible to severe disease, with high attendant rates of mortality [1,2,3,4]. These observations suggested an age-related link between antiviral immunity and disease outcome, but a precise understanding of the determinative biological processes has been lacking to date.

Physiological ageing has been linked with a progressive decline in immune functionality, known as immunosenescence, which could account for the increased vulnerability of elderly individuals to emerging infectious agents, such as SARS-CoV-2 [5,6]. In line with this notion, a relative scarcity of naïve T cells has been associated with severe coronavirus disease 2019 (COVID-19) [7]. Naïve T cells are known to become less frequent and more dysfunctional with age [8,9,10]. It is therefore plausible that functional and numerical alterations in the naïve pool could impair the induction of *de novo* SARS-CoV-2-specific effector/memory T cells in the elderly, consistent with previously described age-related immune deficits in the context of acute COVID-19 [11].

To address this potential mechanistic link, we used an *in vitro* priming approach to quantify the induction of epitope-specific effector/memory CD8^+^ T cells from naïve precursors as a function of age. Importantly, all donors were unvaccinated and previously unexposed to SARS-CoV-2, enabling a robust assessment of priming efficacy in the absence of immunological memory. Our data revealed that advanced age was associated with an impaired ability to mount *de novo* CD8^+^ T cell responses against a range of immunodominant antigens derived from SARS-CoV-2.

## 2. Materials and Methods

### 2.1. Donors

Peripheral blood samples were obtained from anonymized donors (age, 20–69 years; female, n = 2; male, n = 14) via the Blood Bank at Azienda Ospedaliera-Universitaria di Ferrara, Italy. The protocol was approved by the Azienda Unità Sanitaria Locale di Ferrara. All donors expressed human leukocyte antigen (HLA)-A2. Peripheral blood mononuclear cells (PBMCs) were isolated from buffy coats via density gradient centrifugation using Ficoll-Paque (GE Healthcare, Milan, Italy) and cryopreserved in fetal bovine serum (FBS; Euroclone, Milan, Italy) containing 10% dimethyl sulfoxide (DMSO; Sigma-Aldrich, Milan, Italy). Samples were obtained either before or during 2020. Donors in the latter group were only included after testing seronegative for SARS-CoV-2. Serostatus was assessed using a SARS-CoV-2 RBD Total Antibody ELISA Kit (Mabtech, Stockholm, Sweden).

### 2.2. Peptides and Tetramers

HLA-A2-restricted peptides were synthesized in solid phase and purified via high-performance liquid chromatography (>97%). The melanoma-derived EV10 peptide (ELAGIGILTV) was used as a model antigen, given constitutively high cognate precursor frequencies among individuals expressing HLA-A2 [12]. SARS-CoV-2 peptides were selected from different viral proteins on the basis of immunodominance/immunoprevalence [13] and matched to the wildtype strain HKU-001a (Table 1).

Peptides were dissolved initially in DMSO and used at a final concentration of 1 μM. Fluorescent peptide/HLA-A2 tetramers were generated as described previously [14]. All tetramers were used at an identical titer immediately after conjugation to ExtrAvidin-PE (Sigma-Aldrich).

### 2.3. In Vitro Priming of Human Antigen-Specific CD8^+^ T Cells

PBMCs were thawed and resuspended in 75-cm^2^ tissue culture flasks (7 × 10^7^ cells/flask to prime SARS-CoV-2-specific naïve T cells) or 48-well tissue culture plates (2.5 × 10^5^ cells/well to prime EV10-specific naïve T cells) containing AIM-V medium (Thermo Fisher Scientific, Monza, Italy) supplemented with FLT3L (50 ng/mL; Miltenyi Biotec, Bologna, Italy) to mobilize resident dendritic cells (DCs) [13]. After 24 h (day 1), the corresponding peptides were added to the cultures at a final concentration of 1 μM, and DC maturation was induced with TNF-α (1000 U/mL; Miltenyi Biotec), IL-1β (10 ng/mL; Miltenyi Biotec), IL-7 (0.5 ng/mL; R&D Systems, Minneapolis, MN, USA), and prostaglandin E2 (1 μM; Calbiochem, Milan, Italy). On day 2, complement-inactivated FBS (Euroclone) was added at a final *v*/*v* ratio of 10%. Medium was replaced on days 4 and 7 with fresh RPMI 1640 (Euroclone) enriched with 10% FBS (Euroclone), non-essential amino acids (1X; Euroclone), and sodium pyruvate (1 mM; Sigma-Aldrich). Antigen-specific CD8^+^ T cells were characterized on day 10.

### 2.4. ELISpot Assay

Functional responses were quantified using a human interferon (IFN)-γ ELISpot PLUS (HRP) Kit (Mabtech). PBMCs were primed as described above and seeded in duplicate at 1 × 10^5^ cells/well in 100 μL of fresh RPMI 1640 (Euroclone) enriched with 10% FBS (Euroclone), non-essential amino acids (1X; Euroclone), and sodium pyruvate (1 mM; Sigma-Aldrich). Experimental wells contained peptides at a final concentration of 1 μM. Negative control wells lacked peptide, and positive control wells contained anti-CD3 (clone CD3-2; Mabtech). Plates were incubated for 24 h and developed according to the manufacturer’s instructions (Mabtech). IFN-γ-secreting cells were quantified as spot-forming units (SFUs) per 10^6^ cells using an automated ELISpot Reader (AELVIS, Hannover, Germany). Positive responses were assigned after background subtraction at 50 SFUs per 10^6^ cells (mean). Data were excluded in the absence of a response to anti-CD3.

### 2.5. Flow Cytometry

*Ex vivo* analyses were performed using thawed PBMCs. Cells were stained first with LIVE/DEAD Fixable Aqua (Thermo Fisher Scientific) for 10 min at room temperature and then with the following directly conjugated monoclonal antibodies for 15 min at room temperature: anti-CCR7–PE-Cy7 (clone 3D12; BD Biosciences, Milan, Italy), anti-CD8–APC-Cy7 (clone SK1; BD Biosciences), anti-CD27–FITC (clone M-T271; Miltenyi Biotec), anti-CD45RA–PerCP-Cy5.5 (clone HI100; Thermo Fisher Scientific), anti-HLA-DR–APC (clone LN3; Thermo Fisher Scientific), and anti-PD1–PE (clone REA1165; Miltenyi Biotec). Naïve T cells were defined as CCR7^+^CD27^+^CD45RA^+^.

Tetramer staining was performed using 2–3 × 10^6^ primed cells on day 10. Cells were stained serially with LIVE/DEAD Fixable Aqua (Thermo Fisher Scientific) for 10 min at room temperature, individual or pooled tetramers for 15 min at 37 °C, and the following directly conjugated monoclonal antibodies for 15 min at room temperature: anti-CD3–PerCP-Cy5.5 (clone SK7; Thermo Fisher Scientific), anti-CD4–FITC (clone RPA-T4; BD Biosciences), anti-CD8–APC/Fire 750 (clone RPA-T8; BioLegend, Amsterdam, Netherlands), anti-CD14–FITC (clone 18D11; ImmunoTools, Friesoythe, Germany), and anti-CD19–FITC (clone HI19a; ImmunoTools). After fixation/permeabilization using a Transcription Factor Buffer Set (BD Biosciences), cells were stained with anti-Tbet–eFluor 660 (clone 4B10; Thermo Fisher Scientific) for 40 min at 4 °C. Tetramer-labeled CD3^+^CD8^+^ T cells were identified among viable cells after exclusion of FITC^+^ events (dump channel). A positivity threshold was calculated after performing identical experiments in donors lacking expression of HLA-A2.

Data were acquired using a FACSCanto II (BD Biosciences), compensated using CompBeads (BD Biosciences), and analyzed using FlowJo software version 10.8 (FlowJo LLC, Ashland, OR, USA).

### 2.6. HLA–Peptide Binding Assay

Aliquots of 1 × 10^6^ T2 cells, a lymphoma-derived line expressing low amounts of HLA-A2, were incubated overnight at 26 °C in 1 mL of AIM-V medium containing individual peptides at concentrations ranging from 10^−5^ M to 10^−8^ M. Cells were then incubated for a further 4 h at 37 °C, washed extensively in phosphate-buffered saline, and stained with anti-HLA-A2–PE (clone BB7.2; BioLegend). Mean fluorescence intensity was determined via flow cytometry and used to calculate peptide-induced stabilization as the percent increase over baseline expression of HLA-A2 [15,16].

### 2.7. Statistics

Statistical significance was assessed using the Mann–Whitney U test or the Pearson correlation coefficient in Prism software version 8 (GraphPad, San Diego, CA, USA).

## 3. Results

### 3.1. Cohort Description

PBMCs were collected from anonymized adult donors before (2018–2019) or during 2020 (Figure 1a). Donors in the latter group were only included after testing seronegative for SARS-CoV-2. Samples were stratified according to age as “young” (range, 20–52 years; median, 25 years; female, n = 2; male, n = 6) or “old” (range, 65–69 years; median, 68 years; female, n = 0; male, n = 8).

Basal levels of activation and differentiation in the peripheral CD8^+^ T cell compartment were characterized using flow cytometry. A proportionate reduction in the size of the naïve pool was observed with age (Figure 1b). Moreover, higher fractions of total (including naïve) and memory (excluding naïve) CD8^+^ T cells displayed an activated phenotype in old *versus* young individuals, as defined by the expression frequencies of HLA-DR and PD-1 (Figure 1c).

### 3.2. Peptide Characterization

HLA-A2-restricted peptide epitopes derived from different viral proteins were used to measure primary CD8^+^ T cell responses against SARS-CoV-2 (Table 1). The affinity of each peptide for this restriction element was determined using a surface stabilization assay. All peptides bound strongly to HLA-A2. In most cases, stabilization was detected at low peptide concentrations (10^−7^ M and 10^−8^ M), although some differences were apparent, especially at high peptide concentrations (10^−5^ M and 10^−6^ M), consistent with a binding hierarchy (Figure 2). The lowest relative affinities were observed for the spike-derived peptides YLQ and TLD.

### 3.3. Ageing Impairs the Induction of Effector CD8^+^ T Cells Defined by the Expression of T-bet

PBMCs were stimulated *in vitro* with the model peptide EV10 and a panel of peptides derived from SARS-CoV-2 (Table 1), all restricted by HLA-A2 (Figure 3a). Epitope-specific CD8^+^ T cells were quantified via tetramer staining on day 10. This experimental approach to the measurement of priming efficacy has been fully validated in humans and mice [10,17].

EV10-specific CD8^+^ T cells from young donors expanded to a substantially greater extent than EV10-specific CD8^+^ T cells from old donors, resulting in a median frequency difference that exceeded one order of magnitude (Figure 3b). In contrast, only very low frequencies of SARS-CoV-2-specific CD8^+^ T cells (often <0.1%) were detected using a pool of tetramers (n = 5), irrespective of age (Figure 3b). Similar results were obtained using each tetramer individually (Figure 3c).

To extend these findings, we measured the expression of T-bet, a transcription factor that plays a key role in effector differentiation, among primed CD8^+^ T cells specific for EV10 or SARS-CoV-2. T-bet was expressed at lower frequencies among primed CD8^+^ T cells from old *versus* young donors, irrespective of specificity (Figure 3d). Moreover, a direct correlation was observed between T-bet expression frequencies among EV10-specific CD8^+^ T cells and T-bet expression frequencies among SARS-CoV-2-specific CD8^+^ T cells (Figure 3e).

Collectively, these data suggested that age rather than specificity was a key determinant of effector CD8^+^ T cell differentiation, as defined by the expression of T-bet.

### 3.4. Ageing Impairs the Induction of Effector CD8^+^ T Cells Defined by the Expression of IFN-γ

To assess the functional correlates of differential T-bet expression patterns among epitope-specific CD8^+^ T cells primed via our *in vitro* approach, we used ELISpot assays to quantify the secretion of IFN-γ.

IFN-γ response frequencies were consistently higher among young *versus* old donors after stimulation with peptides derived from SARS-CoV-2 (n = 9) (Figure 4a). In the latter group, no responses were observed against the spike-derived peptides KIA and TLD, and responses against the non-spike-derived peptides SLV, MLD, LLY, and HLV were limited to a single donor in each case (Figure 4b). Accordingly, there was an age-related difference in the frequency of epitope recognition, albeit just below the threshold required for significance (Figure 4c). Of note, T-bet expression frequencies among SARS-CoV-2-specific CD8^+^ T cells correlated directly with epitope recognition frequencies (immunoprevalence), measured as a binary outcome for each peptide (Figure 4d), and overall response frequencies (immunodominance), measured as cumulative SFUs (Figure 4e). No correlation was detected between epitope recognition frequencies and peptide binding affinities for HLA-A2 (Appendix A).

Collectively, these data suggested that age rather than specificity was a key determinant of effector CD8^+^ T cell differentiation, as defined by the induction of IFN-γ.

## 4. Discussion

Ageing is known to alter the composition and functionality of the immune system [8,10,18]. These changes limit the ability of elderly individuals to mount *de novo* cellular immune responses against previously unencountered pathogens in the context of natural infection and/or vaccination [9,19]. T cells are particularly susceptible to the ravages of age, potentially explaining the vulnerability of elderly individuals to emerging viral infections, such as SARS-CoV-2 [2]. This notion has gained traction following descriptions of an inverse correlation between the size of the naïve pool and the severity of COVID-19 [7,20,21].

In this study, we used an *in vitro* approach to prime epitope-specific CD8^+^ T cells from healthy donors stratified by age, ensuring a complete absence of prior exposure to antigens derived from SARS-CoV-2. We found that age was associated with reductions in the diversity and magnitude of functional CD8^+^ T cell responses elicited against SARS-CoV-2, measured via the recall induction of IFN-γ. Of note, we also found that epitope recognition frequencies were not associated with peptide binding affinities for HLA-A2, consistent with previous work [22,23,24].

In young donors, EV10 elicited substantially more robust CD8^+^ T cell responses than any of the peptides derived from SARS-CoV-2, most likely reflecting preservation of the naïve pool and the constitutively high precursor frequencies associated with the expression of HLA-A2 [13]. In contrast, viral specificities were barely detectable via tetramer staining on day 10, irrespective of age. Similar findings have been reported previously [25,26]. SARS-CoV-2-specific responses were nonetheless detected functionally. This discrepancy could reflect lower thresholds for activation *versus* tetramer binding and/or differences in the respective limits of detection [27]. Moreover, the functional data revealed an age-related deficit in the mobilization of effector CD8^+^ T cells against various epitopes derived from SARS-CoV-2, consistent with the lower T-bet expression frequencies observed among tetramer-labeled CD8^+^ T cells from old *versus* young donors at the same time point, an observation that extended to the model antigen EV10.

These collective observations confirmed and extended previous findings based on the use of peptide matrices to assess priming efficacy [13]. In particular, we used a cleaner approach focused on single epitopes to identify functional responses and further incorporated phenotypic assessments to provide additional mechanistic information, namely the expression of HLA-DR, PD-1, and T-bet. Accordingly, we generated a comprehensive dataset showing that age-related changes in the basal activation status and overall size of the naïve T cell pool were linked with an impaired ability to mount *de novo* CD8^+^ T cell responses against previously unencountered pathogens, exemplified in this study by SARS-CoV-2.

Immunosenescence is thought to underlie the vulnerability of elderly individuals to COVID-19 [28]. The data presented here revealed a global decline in priming efficacy with age, which led to a reduction in the diversity of functional CD8^+^ T cell responses elicited *in vitro* against a panel of typically immunodominant epitopes derived from SARS-CoV-2. Although further studies will be required to link these findings with immune control of viral replication and vaccine efficacy, our work has established a testable mechanistic paradigm that could expedite the development of new interventions designed to enhance and/or preserve the operational integrity of the naïve T cell pool as a means of protecting the elderly against potentially fatal episodes of COVID-19.

## Figures and Tables

**Figure 1 vaccines-11-00154-f001:**
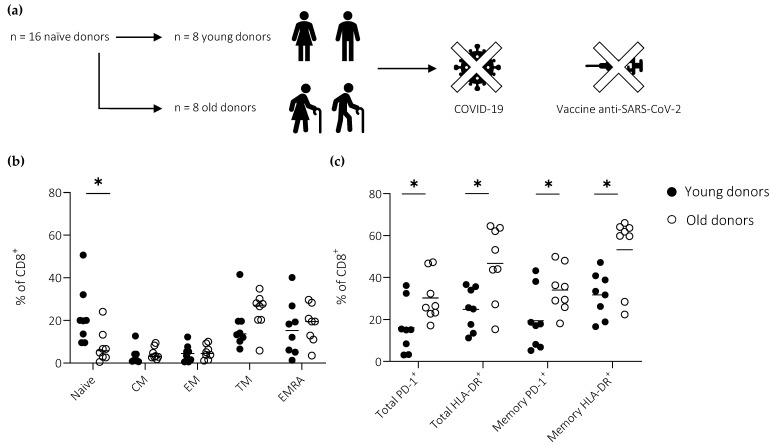
Activation and differentiation in the peripheral CD8^+^ T cell compartment as a function of age. (**a**) Donors were stratified according to age. (**b**) Frequencies of naïve (CCR7^+^CD27^+^CD45RA^+^), central memory (CM; CCR7^+^CD27^+^CD45RA^−^), transitional memory (TM; CCR7^−^CD27^+^CD45RA^−^), effector memory (EM; CCR7^−^CD27^−^CD45RA^−^), and terminally differentiated EM (EMRA; CCR7^−^CD27^−^CD45RA^+^) CD8^+^ T cells. (**c**) Expression frequencies of HLA-DR and PD-1 among total (including naïve) and memory (excluding naïve) CD8^+^ T cells. (**b**,**c**) Horizontal lines represent median values. * *p* < 0.05 (Mann–Whitney U test).

**Figure 2 vaccines-11-00154-f002:**
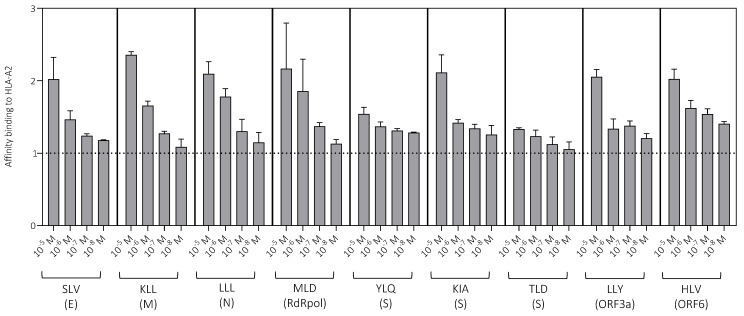
Peptide binding to HLA-A2. Peptide binding affinity was measured across a range of concentrations using a surface stabilization assay. Data are shown as median + SEM. The dotted line indicates no binding (1). E, envelope; M, membrane; ORF, open reading frame; RdRpol, RNA-dependent RNA polymerase; S, spike.

**Figure 3 vaccines-11-00154-f003:**
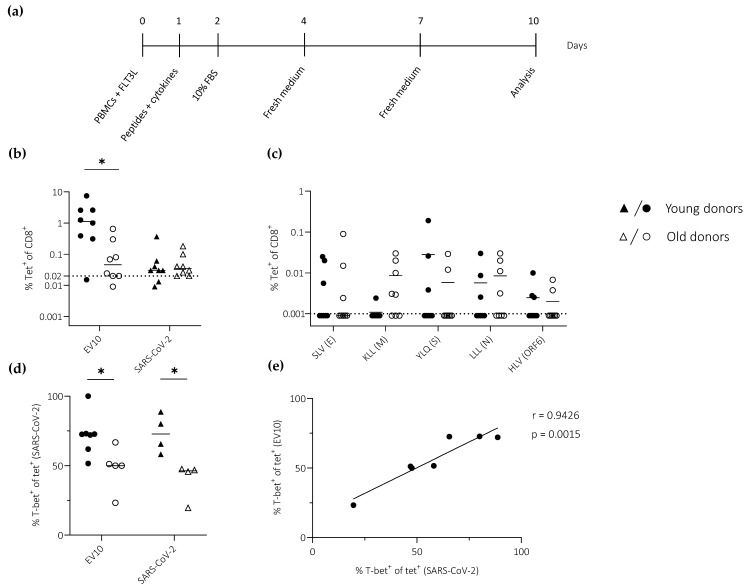
T-bet expression among primed CD8^+^ T cells specific for EV10 or SARS-CoV-2. (**a**) Schematic representation of the *in vitro* priming approach. (**b**,**c**) Epitope-specific CD8^+^ T cell frequencies quantified using the EV10 tetramer or SARS-CoV-2 tetramers in a single pool (n = 5) (**b**) or individually (**c**). Dotted lines indicate the limit of detection. (**d**) T-bet expression frequencies among tetramer-labeled CD8^+^ T cells quantified as in (**b**). (**e**) Pearson correlation showing T-bet expression frequencies among EV10-specific CD8^+^ T cells *versus* T-bet expression frequencies among SARS-CoV-2-specific CD8^+^ T cells. (**b**–**d**) Horizontal lines represent median values. * *p* < 0.05 (Mann–Whitney U test).

**Figure 4 vaccines-11-00154-f004:**
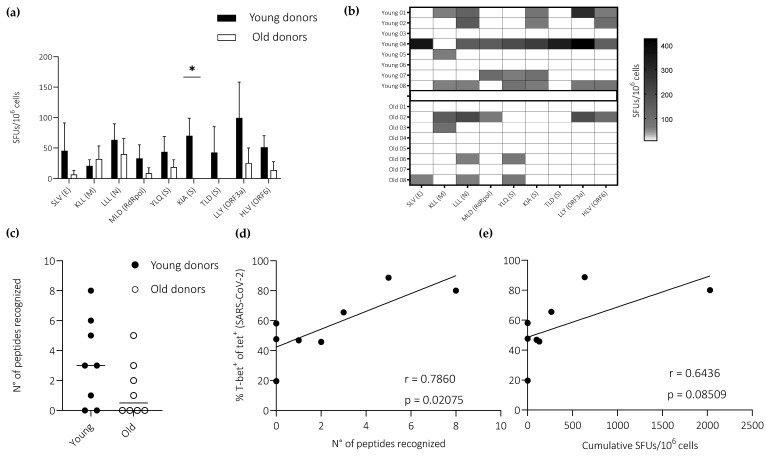
IFN-γ expression among primed CD8^+^ T cells specific for SARS-CoV-2. (**a**) Epitope-specific response frequencies quantified using ELISpot assays to capture IFN-γ after stimulation with peptides derived from SARS-CoV-2. Responses above 50 SFUs per 10^6^ cells were considered positive. Data are shown as median + SEM. (**b**) Data from (**a**) shown as a heatmap. (**c**) The total number of peptides recognized by each donor. Horizontal lines represent median values. (**d**,**e**) Pearson correlations showing T-bet expression frequencies among SARS-CoV-2-specific CD8^+^ T cells *versus* epitope recognition frequencies, measured as a binary outcome for each peptide (**d**), and overall response frequencies, measured as cumulative SFUs (**e**). Dots correspond with the data shown in Figure 3d (young, n = 4; old, n = 4). (**a**) * *p* < 0.05 (Mann–Whitney U test).

**Table 1 vaccines-11-00154-t001:** SARS-CoV-2 peptides and tetramers restricted by HLA-A2.

Protein	Peptide ID	Sequence	Residues	Length	Tetramer
E	SLV	SLVKPSFYV	50–58	9	A*02:01 SV9
M	KLL	KLLEQWNLV	15–23	9	A*02:01 KV9
N	LLL	LLLDRLNQL	222–230	9	A*02:01 LL9
RdRpol	MLD	MLDMYSVML	5291–5299	9	-
S	YLQ	YLQPRTFLL	269–277	9	A*02:01 YL9
S	KIA	KIADYNYKL	417–425	9	-
S	TLD	TLDSKTQSL	109–117	9	-
ORF3a	LLY	LLYDANYFL	139–147	9	-
ORF6	HLV	HLVDFQVTI	3–11	9	A*02:01 HI9

## Data Availability

All original data presented in this study are included in the article/Appendix A. Further inquiries should be directed to the corresponding author (nclfnc1@unife.it).

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
