# Peer review of "Ageing Curtails the Diversity and Functionality of Nascent CD8+ T Cell Responses against SARS-CoV-2"

_vaccines, 2023, doi:10.3390/vaccines11010154_

Round 1

Reviewer 1 Report

The authors experimentally studied T-cell immune responses against SARS-CoV-2. They found that the age was associated with reductions in the diversity and magnitude of functional CD8+ T cell responses elicited against SARS-CoV-2. and that there is no correlation between immunoprevalence and peptide binding affinity for HLA-A2. Although the results may not be so surprising, their approach is scientifically sounds and all the finding in this study was fully supported by the in vitro assays the authors employed. Therefore, I would suggest acceptance of this manuscript as is.

Author Response

We are grateful to the Reviewer for the appreciation of our manuscript

Reviewer 2 Report

This manuscript interestingly tackles the somewhat neglected issue of adaptive cellular immunity in COVID 19 viral infection, through the original angle of aging. A number of tests was used in subjects considered naïve for exposure to SARS-CoV2, thus testing the potential of mounting immune responses against a series of peptides of this virus. Indeed, this was demonstrated to be efficient, with lower levels of response in elderly individuals. In fact, this work looks like an ancillary study of a previous report of the same authors (Frontiers in Immunology, 2021). More emphasis should be brought to the differences between these two papers and how the submitted material extends the previous report that dealt with more patients and more peptides but basically reached the same conclusions.

Here, description of the results is definitely too sketchy. There should be no references in this part of the manuscript but a more detailed and systematic report of the data shown. There is no need to repeat what is already explained in the methods section.

The discussion should focus as mentioned above to the added value of the experiments reported compared to previous work and provide more detailed explanations about the meaning of the findings. The notion of immunoprevalence, allegedly in Figure S1 should be evoked in the results section and properly documented by an explanation of this figure. The conclusion about a “a testable mechanistic paradigm” and how this could lead to “new interventions to protect the elderly” is far too vague and should be expanded

Minor

The country or US State of manufacturers should be indicated at first mention.

A more precise time frame of sampling should be provided.

A few typos should be amended.

Author Response

-“More emphasis should be brought to the differences between these two papers and how the submitted material extends the previous report that dealt with more patients and more peptides but basically reached the same conclusions.”

The Reviewer is right that this aspect is poorly discussed. We have now dedicated one paragraph of the discussion section to this item

-“Here, description of the results is definitely too sketchy. There should be no references in this part of the manuscript but a more detailed and systematic report of the data shown. There is no need to repeat what is already explained in the methods section.”

We thank the reviewer for this suggestion. We have expanded the results description and included Figure S1 in the results section. At the same time, we have removed sentences repeating concepts already mentioned in the method sections or that were discussing the results in the context of current literature.

-“The discussion should focus as mentioned above to the added value of the experiments reported compared to previous work and provide more detailed explanations about the meaning of the findings. “

The current manuscript differentiates from the previous one for the following concepts:

-the use of single epitopes and not matrices for ELISPOT detection (cleaner approach)

-the use of tetramer staining as additional readout, also allowing qualitative measures (e.g. T-bet expression)

-the comparison of primary responses against 2 antigens (SARS-CoV-2 and melanoma)

-the characterization of peptide affinities.

While the last issue was already discussed, the first three improvement are now better contextualized in the new paragraph added to the discussion

-“The notion of immunoprevalence, allegedly in Figure S1 should be evoked in the results section and properly documented by an explanation of this figure.”

We agree and we have now included a description of the Figure in the results section

-“The conclusion about a “a testable mechanistic paradigm” and how this could lead to “new interventions to protect the elderly” is far too vague and should be expanded”

We have now better explained what we meant (preservation of the naïve T-cell numbers/functions)

Minor

-“The country or US State of manufacturers should be indicated at first mention.”

Countries of manufacturer (distribution centres) have been indicated

-“A more precise time frame of sampling should be provided.”

We have now included the recruitment time frame

-“A few typos should be amended.”

A native English speaker has gone through the manuscript and fixed typos

Reviewer 3 Report

Proietto et al. examined the the CD8 T cell response to SARS-CoV-2 as a function of age.  Using cells from human donors there was a clear correlation with the differentiation of precursor cells into CD8 T cells as a function of donor age.  This paper was interesting and clearly presented and worthy of publication.

One additional thing that would be broadly interesting is - if previous vaccination to SARS-CoV-2 improves the CD8 T cell response in older populations of donors. 

Author Response

We are grateful for the positive comments, The Reviewer is right that further studies on vaccinated individuals are required. We have added the following sentenced to the discussion:

“Although further studies will be required to link these findings with early immune control of viral replication and responses to vaccines…”